# Addressing the “Lying Flat” Challenge in China: Incentive Mechanisms for New-Generation Employees through a Moderated Mediation Model

**DOI:** 10.3390/bs14080670

**Published:** 2024-08-02

**Authors:** Jie Zhou, Junqing Yang, Bonoua Faye

**Affiliations:** 1School of Business Administration, Shanxi University of Finance and Economics, Taiyuan 030006, China; zhoujie@sxufe.edu.cn; 2School of Public Administration and Law, Northeast Agricultural University, Harbin 150030, China; bonoua.faye2021@neau.edu.cn

**Keywords:** total rewards, team member proactivity, corporate social responsibility perception, calling

## Abstract

Given the increasing emphasis on teamwork in contemporary organizations and the growing prominence of younger employees in the workplace, it is crucial to encourage their proactivity in navigating complex internal and external environments. Total rewards are a highly effective means of motivating the new generation of employees; however, there is limited research on whether and how total rewards stimulate team member proactivity. To address this objective, this study utilizes survey data (n = 423) and employs hierarchical regression and bootstrap methods. In essence, this paper aims to construct a moderated mediation model to examine the relationship between total rewards and team member proactivity among Chinese new-generation employees (born after 1990). The results indicate that total rewards significantly enhance the team member proactivity of new-generation employees. Furthermore, calling serves as a significant mediator in this relationship. The perception of corporate social responsibility also plays a crucial role, positively moderating the relationship between total rewards and calling. This, in turn, positively influences team member proactivity through the mediation of calling. Accordingly, this research provides valuable insights for managers aiming to effectively engage the new generation of employees and boost team performance. In essence, our model enriches the understanding of how compensation practices can be leveraged to boost proactivity among the new generation of employees.

## 1. Introduction

The Sustainable Development Goals (SDGs) represent a well-thought-out and comprehensive policy agenda designed to create a better world [1]. In parallel, innovation capability naturally becomes a crucial guarantee for businesses to maintain competitiveness and achieve positive performance growth [2]. Therefore, issues related to social reproductive work must be aligned to support and sustain a decent work agenda, particularly in line with the objectives of SDG 8 [3]. This goal focuses on promoting sustained, inclusive, and sustainable economic growth, full and productive employment, and decent work for all. The research on developing effective incentive mechanisms aligns with creating better work environments that enhance productivity and job satisfaction, which are crucial for economic growth and employment quality.

Consequently, employment quality and performance remain challenging, but inclusive leadership can enhance employee motivation and innovative behavior [4]. During the last decade, companies have been increasing their expectations of employees. In other words, employee expectations are shifting in response to the rapid socio-economic developments that are transforming work practices. They expect them to continuously update their skills, identify opportunities for improvement, anticipate and prevent future issues, and be more proactive [5]. As modern organizations increasingly focus on teamwork, the dynamics within teams have led to more interconnected work processes and content. This change requires a rise in proactive behavior oriented toward teamwork, known as team member proactivity. Therefore, team member proactivity reflects the extent to which an individual engages in self-starting, future-directed behavior to change a team’s situation or the way the team works [6]. It has been proven to improve team efficiency, strengthen intra-team cooperation, and create a better working environment [7,8]. However, while leaders have increasingly relied on the proactive contributions of individual employees to solve current and anticipated future problems, to date, theoretical research on team member proactivity has been relatively scarce [9].

Although there is a consensus that workplace proactivity is vital for both employees and organizations, little is known as to whether proactivity is beneficial for employees to develop leadership capabilities [10]. In other words, workplace proactivity and the superior–subordinate relationship play crucial roles in fostering a productive and harmonious work environment. Proactivity among team members entails questioning and potentially altering established norms, which may carry certain challenges. The outcomes of such behavior are not always effective [11]. However, compared to proactive behaviors focused on individual efforts, team member proactivity goes beyond the individual and seeks to enhance the overall effectiveness of the team. As a result, it involves more intricate aspects, such as team system design and adjustments in the tasks of other team members. These changes can create uncertainty among team members and potentially increase the social costs associated with engaging in such behavior [6]. Therefore, compared to individual task proactivity, the factors that affect team member proactivity are varied, and need for more comprehensive and systematic interaction among individual and situational factors. Previous studies, such as those conducted by El Baroudi et al., have systematically reviewed the literature on team member proactivity. These reviews highlight the limited understanding of how individual and situational factors influence team member proactivity [9,12]. They hypothesize that teams that utilize a mixed reward structure are more likely to engage in team-oriented proactive behavior [9]. With the new generation of post-1990s employees gradually becoming the main workforce in China, there has been growing advocacy for the iteration and reform of management methods that align with their diverse values and preferences [13]. Thus, the total rewards—a mixed form of reward that covers both internal and external rewards, flexible and diversified combination forms, and pays more attention to employees’ psychological and emotional states—have become the most powerful motivational tool in the current organizational environment [14]. However, in the workplace, the new generation of employees in China is experiencing a “*lying flat*” mentality and a lack of motivation. The concept of “*Lying flat*” literally has become a buzzword in Chinese web media, which refers to a simple lifestyle without effort [15]. This latter concept is linked to “losing face”, which refers to the concept of losing respect, dignity, or reputation in the eyes of others. Globally, in Chinese culture, “lying flat” (躺平, tǎng píng) and “losing face” (丢脸, diū liǎn) are related concepts that reflect different aspects of societal expectations and individual responses.

However, in this study, we closely focus on the concept of “lying flat”. The behaviors related to “lying flat” indicate that the current implementation of total rewards in the organization is not optimal. In other words, it has recently become an incentive management challenge for organizations to implement a total rewards system [16]. Accordingly, the organization should find ways to effectively motivate and inspire the new generation of employees through total rewards. Addressing this issue is vital for managing human capital, particularly in China companies.

Currently, there is a lack of empirical research on the internal mechanism of total reward on team member proactivity. Previous studies mainly focus on social exchange theory, self-determination theory, resource conservation theory, and other theoretical perspectives [17,18]. However, the specificity of team member proactivity implies that the main strategy to encourage it should involve defining their work roles based on their identity as a team member, aligning their interests with those of the team, and fostering a sense of shared destiny with the team [9]. In this context, social identity theory captures the emotions and values individuals derive from their group membership. Therefore, behaviors like individuals positioning themselves psychologically within a team to gain a sense of belonging and striving to enhance their self-worth are closely aligned with social identity theory [19]. At the same time, such emotion experienced by the individual is also an important mechanism in the formation of calling [20]. In addition, the external resource endowment, which takes into account individual material and spiritual needs, is the strongest incentive for the generation of calling [21,22]. This offers substantial evidence for research on the correlation between total rewards and team member proactivity.

Additionally, social identity theory suggests that internalizing an organization’s reputation enables individuals to gain and enhance positive self-worth through their group membership [23]. Corporate social responsibility is related to the company’s external reputation, while total rewards support employees within the organization. Thus, existing research demonstrates that effectively implementing this external moral framework helps employees identify more with the organization and the team, transforming external success into an internal mission. This significantly influences the development of employees’ sense of calling [24,25].

From then on, this study aims to examine the impact of a total rewards incentive strategy on the team member’s proactive behaviors of the new generation of employees in Chinese enterprises. Those born after 1990 are considered new-generation employees in China. This choice is driven by the context of technological innovations and advancements, such as the implementation of 5G. In this era, young people represent a vital source of national strategic talent, surpassing the previous generation in their potential and capability. Furthermore, this study specifically explores the role of calling and perception of corporate social responsibility as mediators and moderators. Consequently, this research contributed to the development of effective incentive mechanisms that align with creating better work environments that enhance productivity and job satisfaction, which are crucial for economic growth and employment quality.

## 2. Literature Review and Research Hypothesis

### 2.1. Total Rewards and Team Member Proactivity

Proactive behavior is defined as “taking the initiative to improve current situations or create new ones. It involves challenging the status quo rather than passively adapting to current conditions [26]”. Subsequently, researchers expanded the definition to encompass more than just individual actions, recognizing that employees’ work behaviors are not solely representative of themselves but are also influenced by the social contexts of their work roles. For instance, employees are integral to their work or organizational units, and their behaviors can extend beyond their roles, impacting and being shaped by various levels within the organization. Employees may align their actions with their objectives, those of their colleagues, or broader organizational goals [8]. Griffin et al. (2007) and other scholars have been pivotal in researching this issue in the field [6]. They introduced the concept of team member proactivity, which is defined by the degree to which an individual initiates self-directed, future-oriented actions aimed at altering a team’s conditions or operational methods. For instance, nurses might propose modifications to the team’s shift schedule as a form of proactive behavior [6]. Unlike team proactivity, which represents a group’s collective thought and capabilities, team member proactivity is an individual variable. It showcases the personal initiative of each team member to engage in proactive behavior independently rather than as part of a collaborative team effort [9].

Compensation incentives are crucial tools for organizations to motivate employees and manage their performance. Although this topic is often underrepresented in theoretical discussions, it receives considerable attention in practical applications. Since the 19th century, compensation practices have undergone substantial evolution, transitioning from incentives like salaries, bonuses, and equity to the 20th century’s introduction of rewards such as leadership recognition and internal promotions during the knowledge era. This evolution culminated in the adoption of total compensation strategies by prominent enterprises in developed countries in the 1970s. This evolution mirrors the changing needs of employees, from being viewed as “*economic men*” focused on financial rewards to “*knowledge men*” valuing intellectual recognition to “*self-actualizing men*” seeking personal and professional fulfillment.

Consequently, compensation strategies have transitioned from purely external economic rewards to include internal non-economic rewards, culminating in a holistic approach encompassing both internal and external rewards, known as total rewards [27]. In the early 2000s, WorldatWork provided a widely embraced definition of total rewards as “*whatever an employee considers valuable in an employment relationship*”. This led to the development of a comprehensive five-dimensional model encompassing compensation, benefits, work–life balance, performance, and recognition, as well as development and career opportunities. Over time, companies began to align the design of total rewards incentives with their strategic organizational goals. Recently, this approach has evolved into a strategic model that includes elements such as compensation, work happiness, well-being, modern workplace, and recognition aimed at enhancing the overall performance of the organization [28,29]. As globalization progresses and a new generation becomes the predominant workforce, Chinese enterprises have increasingly recognized the importance of adopting a total remuneration design system that prioritizes employee growth and psychological well-being as a primary means of motivation.

Consequently, Chinese scholars have started to investigate the structural dimensions of total rewards within the Chinese organizational context, examining their effects on employee attitudes and behaviors. This research has contributed to the localization and adaptation of total rewards strategies, tailoring them to meet better the specific needs and cultural nuances of Chinese workplaces [30]. To cater more effectively to the preferences and characteristics of the new generation of Chinese employees, this study employs a scale known for its comprehensive content and thorough qualitative research process. This scale encompasses various dimensions, including compensation security, compensation fairness, work burden, work experience, employee care, career development, and personal value perception [31]. These dimensions can collectively encompass the various aspects of total rewards, providing a holistic view of the rewards system.

Traditional management theories, such as the induction contribution theory, social exchange theory, and expectancy theory, all suggest that employees will act in ways that support organizational goals if they receive appropriate incentives or rewards. The magnitude of these incentives or rewards directly influences the extent to which employees contribute or engage in behaviors that benefit the organization [32]. Proactive behavior, distinct from general organizational citizenship behavior and extra-role behavior, involves transformative and forward-looking actions taken by individuals that go beyond the routine demands of their jobs. The goal of these actions is to effect change in the current environment or within themselves [33]. The proactive behavior of team members, which encompasses aspects like teamwork processes and colleagues’ roles and methods, often entails a significant social cost. Therefore, to encourage employees to engage in such initiatives voluntarily, it is essential to provide a comprehensive and substantial set of incentives and rewards [9]. Total rewards encompass both economic incentives and material benefits, such as compensation security and fairness, as well as developmental incentives and non-material benefits, including work experience, employee care, career development, and personal value perception. The new generation, distinct in their upbringing, work values, and knowledge base, exhibits a profound shift in needs compared to previous generations. Particularly, this generation demands a holistic, coordinated, and balanced approach to meeting both their material and spiritual needs [34]. Effective implementation of a comprehensive total rewards strategy by organizations leads to employees acknowledging the fulfillment of their diverse needs, thereby cultivating a proactive psychological state. Simultaneously, this leads them to make optimistic assessments about the outcomes of their proactive behaviors. As a result, they experience positive emotions and increased energy, which helps mitigate the sense of self-depletion that can arise from the uncertainty of behavior outcomes. Compared to rewards based solely on individual outcomes, this mixed rewards approach is more effective in encouraging the new generation of employees to transcend their roles. It motivates them to concentrate on enhancing team efficiency, actively engage in team interactions and behaviors, and diligently work to meet their collective team responsibilities. In essence, employees who perceive a comprehensive total rewards strategy, which includes both economic and non-economic incentives, are more likely to exhibit proactive behavior aimed at improving team conditions and operational methods. Therefore, our hypothesis suggests the following:

**Hypothesis 1** **(H1):**
*Total rewards can positively influence the proactive behavior of new-generation team members.*


### 2.2. The Mediating Role of Calling

Career calling is defined as a meaningful and continuous experience that drives people to perform their jobs passionately without expecting any extra material reward [35]. The concept of calling origin in religious theology emphasizes divine mandates and placing individuals in specific clerical roles [36]. Subsequently, two interpretations of calling emerged. The first was a neoclassical definition that extended the traditional religious or external summons, characterized as “*a transcendental call beyond oneself*”. This perspective views the purpose of work as serving others, emphasizing a sense of destiny or prosocial responsibility, and highlighting external demands related to duty, obligation, and altruistic values [37]. Furthermore, the other definition, which focuses on the individual, defines it as an intense, meaningful passion that the individual experiences in a particular professional field [35]. It believes that the summoner comes from within and that the purpose of work is for oneself, which is a call for internal preference and self-realization and emphasizes the inner need for enjoyment and satisfaction brought by a professional passion [38]. In this context, Maslow argues that the self-actualized person’s calling is not about internal (“*I want*”) or external (“*I must*”) independent needs but about the match of one with the other. In addition to this compatibility, an individual experiences a feeling of balance, assurance, or fate; the most optimal summoning encounter occurs when the alignment is perfectly suitable [20].

According to the evolving concept, calling is likely to serve as a bridge between the overall incentive of the organization’s total rewards and the proactivity of the one hand, the social identity theory states that when individuals perceive the support and value of the group, they enhance their feelings and significance of staying in the group, form a group identity, and subsequently generate a sense of calling [39]. Conversely, when employees perceive that their career calling is unable to be pursued, they will develop an intention to leave their organizations [40]. Existing studies have shown that external resource endowment, which includes material support for individuals and spiritual needs such as perception of work meaning, self-worth, and social value experienced by individuals in their occupation, is the strongest incentive to form callings. In contrast, internal resource endowment such as occupational ability, work well, and psychological experience can also give individuals strong psychological and behavioral abilities. Driving individuals to pursue their spiritual ideal selves actively helps enhance their calling [41]. The total rewards are a range of rewards that the organization receives for the efforts of its employees, and they serve as a tangible representation of its support. At the same time, the total rewards can form a collection of internal and external resource endowments by satisfying employees’ diversified expectations of work [8]. Therefore, when organizations and employee teams implement overall incentives such as total rewards, employees are most likely to achieve a high level of calling. Based on the arguments above, we have formulated the following hypothesis:

**Hypothesis 2** **(H2):**
*Effective implementation of comprehensive total rewards strategies by organizations can positively influence employees’ sense of calling of the new-generation employees.*


Conversely, calling can have a significant impact on the proactivity of an employee’s team members. According to social identity theory, the dual motivations of minimizing uncertainty and enhancing identity, including bolstering self-esteem and reputation, can bolster an individual’s sense of belonging to the group. This aids in maintaining a positive outlook and behavior toward the group, motivating individuals to strive toward the group’s objectives for personal growth and improvement [19]. Employees who possess a strong sense of calling appear to recognize the importance of social value and social responsibility. The external embodiment of this responsibility and sociality will help team members gain a sense of belonging and security, greatly enhance the collective identity, and reduce the risks and challenges brought by the uncertainty of identity. Therefore, through the first motivation mechanism, employees will engage in prosocial behaviors that contribute to the well-being of others and enhance team performance [8,42]. The modernist perspective of the call emphasizes the sense of meaning and purpose of work, as well as passion, enjoyment, and realization of self-worth. Employees who possess a strong sense of calling will dedicate themselves to their work with greater passion, thereby expanding their psychological, intellectual, and physiological resources, enhancing their sense of self-efficacy, stimulating a more positive emotional experience, and ultimately enhancing their internal motivation to implement team member proactivity through the second motivation: identity promotion. Given the reasoning above, we have developed the subsequent hypothesis:

**Hypothesis 3** **(H3):**
*The new generation of employees with a strong sense of calling are more likely to engage in proactive behaviors within their teams.*


Employees often find themselves working in organizations where their jobs have multiple dimensions [43], requiring strong management and communication. According to social identity theory, the organization’s compensation incentive practice has the potential to enhance communication. When the organization implements the total rewards incentive, individual employees can perceive the group support and the collection of internal and external resources provided by the organization. Such signals stimulate employees’ sense of identity, belonging, and self-esteem as group members and enhance their calling. The internal and external demand experience of the calling helps team members form a positive team identity through the dual motivation of uncertainty reduction and identity promotion, and it encourages members to actively participate in the behavior that is conducive to the team. In a nutshell, organizational implementation of total rewards incentives positively influences employee communication within teams. Therefore, based on this context, we proposed the following:

**Hypothesis 4** **(H4):**
*The sense of calling mediates the relationship between the total rewards and the team member proactivity of the new generation of employees.*


### 2.3. The Moderating Effect of Corporate Social Responsibility Perception

Employee performance shapes the company’s output, so human resource corporate social responsibility (CSR) training is required [1]. CSR generally refers to the impact of corporate strategy and operational practices on the well-being of its major external stakeholders. Due to extensive empirical evidence, it is widely accepted that stakeholders’ perceptions of CSR activities can enhance a company’s reputation [44]. In other words, this paper supports the view that stakeholders’ perceptions of CSR activities can positively influence a company’s reputation. It extends this idea by exploring how specific CSR initiatives and stakeholder engagement strategies contribute to building and maintaining a favorable public image. However, employees may not be able to accurately and objectively understand the specific social responsibility behaviors of enterprises. What really influences employees’ attitudes and behaviors is their psychological perception and overall judgment of CSR activities. The new generation of employees is increasingly focusing on corporate responsibility and social responsibility, and they have set requirements [25] for the company to fulfill social responsibility and promote sustainable development, with the aim of realizing their self-worth and making a positive contribution to society through their work. According to social identity theory, people acquire an individually defined significant identity through the organizations they work for, and they are particularly motivated to identify with organizations that enhance their sense of self-worth. As a representation of the external image and identity of enterprises, CSR shows the social image of enterprises concerned about social welfare, which can greatly enhance the external honor perception of employees.

The values of moral responsibility and benevolent beliefs conveyed by the enterprise, when perceived by the new generation of employees as a better fulfillment of corporate social responsibility, will resonate with them. They will believe that the enterprise is committed to taking responsibility for the community, consumers, and environment, thereby fostering a positive group identity among employees [45]. At this time, the combination of high internal total rewards and a positive external perception of corporate reputation will enhance employees’ sense of trust and identity in the organization from both internal and external aspects. This all-encompassing view not only strengthens the employees’ sense of accountability and camaraderie but also views the organization’s reputation as a recognition of their contributions, thereby fostering a sense of superiority within the organization. They are more likely to feel the meaning, vitality, and enthusiasm of work. During this period, the calling strengthens both internal and external needs [46], thereby enhancing the dual identity motivation of belonging and self-improvement among employees. This, in turn, enables them to set goals that align with the team and encourages members to participate in group activities positively [10]. However, when employees perceive CSR as low or solely symbolic, it challenges the enterprise’s legitimacy. Even if the organization offers high total rewards, the role of moral self-perception among employees can reduce their intrinsic motivation and lead to negative behaviors.

Further, Tajfel and Turner argues that social identity consists of three components: cognition, evaluation, and emotion, which reflect a person’s knowledge, values, and emotional attachment. Among these, the emotional component is the most influential in determining in-group preference. Consequently, this paper focuses on the emotional component of identification, noting that participants have a clear understanding of their team membership due to their assigned roles [47,48]. They further proposed that once social identity is established, individuals are motivated to contribute to the group’s success, demonstrating civic behavior that supports the group’s processes and outcomes.

According to social identity theory, when employees perceive that CSR initiatives are well implemented, it enhances their sense of belonging and organizational pride, strengthening their team identity [19]. Employees see the positive impact of CSR on the company’s reputation, which in turn boosts their self-esteem and fosters a stronger team identity. When these two aspects of identification interact with the internal affirmation of total rewards, employees feel valued both internally and externally. This sense of mutual benefit between employers and employees enhances their sense of belonging and self-worth.

The development and reinforcement of this emotional connection inspire employees to see themselves as “one” with the team and the organization. This unity encourages them to engage in behaviors aligned with their identity goals and motivates them to take proactive steps to contribute to the team’s and the organization’s success [45].

In summary, the interaction of total rewards and the perception of CSR can enhance employees’ sense of calling and stimulate the proactivity of team members through the sense of calling, creating a cycle of motivation and engagement that benefits both the employees and the organization (Figure 1). Consequently, our hypothesis proposes the following:

**Hypothesis 5** **(H5):**
*CSR perception moderates the positive relationship between total rewards and calling, such that the positive relationship is enhanced when CSR perception is high rather than low.*


**Hypothesis 6** **(H6):**
*The indirect effect of total rewards on team member proactivity of the new-generation employees via calling is moderated by CSR perception, such that the effect is enhanced when CSR perception is high rather than low.*


## 3. Materials and Methods

### 3.1. Sample and Procedure

Questionnaires and social surveys are widely used tools in empirical social science [49]. The data used in this study were collected from various sample sources. They focus on the new generation of employees working in technology, research and development, and management at 85 companies located in 10 provinces in China. These companies are found in a variety of domains, including financial, construction, software and technical services, and information transmission.

To ensure the quality of the survey, a matching questionnaire regarding the leaders and employees was used, employing a two-point scoring system. The data were collected over two months. The first step of the questionnaire was related to employees’ self-evaluating the total rewards, CSR perception, proactive personality, and basic personal information. A month later, the second step involves gathering employees’ calling and evaluating the relationship between superiors and subordinates.

A contact person was determined in the proposed survey enterprise, responsible for matching the number of leaders and employees, preparing sealed envelopes, on-site distribution, recycling, and mailing questionnaires. Specifically, we collected 138 questionnaires from leaders and 423 from employees. Therefore, the data screening process plays a crucial role in eliminating questionnaires that lack logical reasoning [50]. As a result, the data screening process revealed that about 35 surveys of leaders and 95 surveys of employees were excluded because they lacked logical coherence or incompleted filling. For instance, if there is a significant discrepancy in responses to similar measurement items before and after, it is considered a lack of logical consistency. Samples exhibiting such discrepancies will be excluded from the analysis. Hence, a total of 103 leaders, matching 328 from employees’ questionnaires, were utilized in this study, accounting for 77.54% of the initial number of questionnaires collected. At the same time, the direct leader evaluates the team member’s proactivity of the employee and fills in the personal information of the leader.

As shown in Table 1 and Table 2, the characteristics of the respondents reveal several differences, with the post-1990 generation representing approximately 43%. The survey indicated that approximately 46.95% of the respondents were male, while 52.44% were female. Furthermore, their educational background showed that 23.17% were attending college, 58.84% were undergraduate students, and 17.99% were master’s students.

### 3.2. Measures

The remaining variables in this study are mature scales commonly used in the classic foreign literature, with the exception of the measurement of total rewards, which is a locally developed Chinese scale. The entries are scored on a Likert 6-point scale with a range of 1 representing “*very inconsistent*” and 6 indicating “*very consistent*”.

#### 3.2.1. Total Rewards

The variables concerning total rewards are measured using a 33-item scale developed by Jin Weizei and Yang Junqing (2022), tailored to the Chinese organizational context and encompassing seven dimensions [31]. Sample items involve measurement sentences such as “*The company provided me with effective training*”, “*My performance bonus is linked to my work contribution*”, etc. Accordingly, the scale’s Cronbach’s α value was 0.954, and the KMO value was 0.938, with all items showing significance at the 0.001 level. Seven eigenvalues corresponding to seven dimensions were extracted, with the cumulative variance interpretation rate reaching 73.41%. The factor loadings of the component matrix after rotation were generally above 0.7, further confirming the scale’s reliability and validity.

#### 3.2.2. Calling

Calling is an individual’s ongoing, meaningful passion for a particular occupation. The calling variable is based on the 12-item scale developed by Dobrow and Tosti-Kharas (2011) [35]. The measurement items include considerations such as “*Without my commitment to my work, my existence would be considerably less meaningful*, etc.”. As a result, this scale had a Cronbach’s α value of 0.919.

#### 3.2.3. Corporate Social Responsibility Perception

The corporate social responsibility perception variable adopts the 10-item scale compiled by Farooq et al. (2014) [51]. In this section, the items also involve several considerations. For instance, the consideration “*I can perceive that our enterprise provides comprehensive and accurate information when supplying goods and services to consumers*, etc., was among the selected items”. Therefore, Cronbach’s α value measurement for this scale was significant, reaching 0.879.

#### 3.2.4. Team Member Proactivity

For the team member proactivity variable, Griffin et al.’s (2007) 3-item scale was used to conduct a leadership assessment [6], with example items such as “*The employee proposed ways to improve teamwork efficiency, etc.*”. Accordingly, compared to the previously selected items, the Cronbach’s α value of team member proactivity for the scale was significant at 0.887.

### 3.3. Control Variables

Three classes—demographics, the worker’s relationship, and individual characteristics—comprise the eight control variables this paper selects. The first concerns the demographic characteristics of employees, including seniority, which reflects the extent of knowledge accumulated over the years, similar to educational attainment. This study also considers the employee’s grade and level of responsibility, which are significant aspects within the company, hence their inclusion as variables. The second control variable addresses social approval, focusing on the duration of the working relationship between leaders and employees, as well as the dynamics between superiors and subordinates, which can influence employee behavior and feelings, as well as how employees perceive their leaders. Law et al. (2000) developed a 6-item scale to specifically measure the relationship between superiors and subordinates [52]. The example item is “When there are differing opinions, I stand by the leader”. As a result, this scale’s Cronbach’s α value reaches 0.766. The third type is the proactive personality. We used the four items with the highest factor load in Bateman and Crant’s (1993) scale to measure this tendentious antecedent variable of active behavior [53]. In this context, the example items include, “*Regardless of how the probability of success is, as long as I believe, I will do it*”. Therefore, this scale’s Cronbach’s α value was 0.773.

## 4. Results

### 4.1. Common Method Bias Test and Confirmatory Factor Analysis

First, based on Harman’s single-factor test, it was found that the first factor accounted for 27.41% of the variance, which is significantly lower than the 40% threshold. Following that, the validity of discrimination was determined. Therefore, as evidenced by Table 3, the overall fit of the original model’s four factors is the most significant and best (*χ*^2^ = 3172.068, *df* = 1945, *χ*^2^*/df* = 1.631, *RMSEA* = 0.044, *CFI* = 0.915, *TLI* = 0.909, *IFI* = 0.916). Based on the four-factor structure, we constructed a five-factor model containing the common method variation factor. Still, the fitting effect did not significantly improve (*RMSEA* = 0.003, *CFI* = 0.014, *TLI* = 0.013, *IFI* = 0.014), further verifying the model’s common method. The deviation is not significant.

### 4.2. Descriptive Statistics and Correlation Analysis

As can be seen from Table 4, there is a significant positive correlation between total rewards and calling (r = 0.398), on the other hand, and a significant positive correlation between total rewards and team member proactivity (r = 0.207), on the other hand with *p* < 0.01. Additionally, there is a significant positive correlation between calling and team member proactivity (r = 0.212; *p* < 0.01. According to the double judgment criteria, the R-value is lower than 0.75, and the variance inflation factor VIF of the core variable is both less than 2 and lower than 10. It is proved that there is no multicollinearity problem in this study. The relevant hypothesis has been preliminarily verified.

### 4.3. Hypothesis Testing

#### 4.3.1. Main Effect and Intermediate Effect Test

Table 5 presents the methodology and outcomes of hierarchical regression analysis. According to Model 6, total rewards have a significant positive effect on team member proactivity (β = 0.24, *p* < 0.01). Accordingly, these results suggest that H1 is supported. In other words, total rewards are positively related to the team member proactivity of the new generation of employees.

In addition, the intermediate effect test shows similar results. As can be seen from Models 2 and 7 in Table 5, after calling is added, the total rewards have a significant positive relationship with calling (β1 = 0.358, *p* < 0.001). Furthermore, calling also has a significant positive relationship with team member proactivity (β2 = 0.165; *p* < 0.05), but the main effect was still significant. However, the coefficient decreased from 0.24 to 0.181, indicating that calling had a partial mediating effect between total rewards and team member proactivity. Consequently, the results support our hypotheses, specifically H2, H3, and H4.

#### 4.3.2. Adjustment Effect Test

To prevent collinearity issues associated with testing interaction terms, cross-multiplicative terms involving total rewards, calling, and CSR perception were constructed through centering and subsequently incorporated into the regression analysis. As can be seen from Model 3 in Table 3, there is a significant interaction between total rewards and CSR perception (β = 0.154, *p* < 0.05).

To more clearly and intuitively illustrate the influence of moderating variables, this study created a two-dimensional interaction graph based on the mean plus or minus one standard deviation and conducted a simple slope comparison analysis. Figure 2 presents this two-dimensional interaction diagram. When the level of CSR perception by employees is higher (+1 SD), the positive impact of total rewards on calling is more pronounced (β = 0.364, *p* < 0.001).

Conversely, when the level of CSR perception by employees is lower (−1 SD), the relationship between total rewards and calling is not significant (β = 0.138, n.s.). Therefore, these results supported our H5. This means that CSR perception positively moderates the relationship between total rewards and calling.

#### 4.3.3. Moderating Mediating Effect

Using the Process macro (Model 7), the bootstrap (5000 samples) method tested whether the 95% confidence interval for each moderator at various levels included zero to assess the significance of the conditional process. The results, displayed in Table 6, show that the mediating effect of calling on team member proactivity is significant when CSR perception is higher than 1. Conversely, this mediating effect is not significant when CSR perception is below 1.

## 5. Discussion

As teamwork becomes increasingly crucial in the workplace, leaders are more reliant on the proactive contributions of individual team members to enhance team performance in addressing both current and predicted future challenges. Despite this, theoretical research on stimulating team member proactivity is scarce. Using survey data, our study found evidence that total rewards serve as the most effective means of motivating new-generation employees, significantly enhancing their team member proactivity. Drawing on social identity theory, we observed that this incentive mechanism operates by reducing uncertainty (thus reinforcing group identity) and enhancing identity value (through meaning construction), thereby intensifying employees’ sense of calling. Additionally, we identified a moderating effect of corporate social responsibility perception. When the internal incentives provided by total rewards align with the external reputation garnered through corporate social responsibility perceptions, it elevates the level of employees’ calling, which in turn amplifies the positive impact on team member proactivity. These results allow us to discuss the theoretical and practical implications.

### 5.1. Theoretical Implications

This study centers on the new generation of employees, emphasizing the significance of examining total rewards theory from the perspective of fulfilling their needs. It explores how total rewards influence employee proactive behavior, thereby addressing the academic community’s call to enhance the study of total rewards theory to a certain degree [13,14]. This study contributes to expanding the theoretical understanding of total rewards. Currently, as the new generation increasingly becomes the primary focus of human resources research, numerous enterprises are beginning to integrate various reward methods to develop new reward systems aimed at better motivating and retaining employees. Practically, most organizations implement strategies of total rewards and comprehensive incentives to varying degrees. However, both theoretical and empirical research on total rewards remains lacking, leading to a noticeable disconnect between compensation theory and compensation practice [14]. Globally, in essence, the exchange of employee effort for employer rewards is central to the employment relationship. Employees consider the overall mix of compensation and benefits, prompting organizations to adjust these packages to attract and retain talent, especially among younger employees. While business managers emphasize total rewards for competitiveness, academic research lags, often focusing separately on compensation or benefits. This gap has led to calls for more comprehensive academic research on total rewards, particularly in China, where theoretical work on the topic is limited.

Secondly, this study delves into the “*black box*” of team member proactivity research. Being proactive is a self-initiated and future-oriented type of motivational work behavior [53,54]. Proactive work behaviors can be self-directed, such as job crafting and seeking feedback, or they can include broader actions like transformative behavior. While proactivity is often seen as a process involving relatively autonomous individuals, organizations increasingly depend on proactive employees who collaborate within teams, relying on each other to accomplish tasks [12]. Therefore, focusing on the team as a whole rather than just individual initiative becomes more important. This reflects the current reality.

In reality, not all employees are equally proactive; some may “*lie flat*”, willing to enhance their personal work efficiency but reluctant to suggest improvements for team efficiency. Theoretical studies to date have primarily focused on the comprehensive measurement of general proactive behavior or specific forms of individual proactive behavior, such as proactive change behavior and job crafting, from the content perspective of proactive behavior. However, research on team-oriented proactive behavior, which classifies proactive actions based on their focus on change and goals, remains scant. As contemporary organizations increasingly emphasize teamwork and individuals frequently collaborate in teams, understanding the factors and mechanisms influencing team member proactivity becomes crucial. However, there is limited research on team-oriented proactive behavior, which categorizes proactive actions based on their focus on change and goals. This study examines the mechanisms of team member proactivity from the perspective of total rewards—an organizational compensation variable—answering the academic call for more nuanced research into the classification of employees’ proactive behaviors [9,26].

Third, leveraging social identity theory, this paper unveils the motivational mechanism by which total rewards encourage team members’ proactivity through the lens of calling. This approach offers a fresh theoretical perspective on stimulating employee initiative. Previous studies that looked at how proactive behavior starts mostly looked at how it works on the inside, using the active motivation model and the social exchange theory [5]. However, prior studies have suggested that both the comprehensive incentive nature of total rewards and the complexity and uncertainty of team member proactivity necessitate a deeper, intrinsic motivational process for a complete explanation. This study proposes that self-driven motivation, which emerges from the integration and alignment of internal and external needs as described by social identity theory’s concepts of uncertainty reduction, identity value enhancement, and calling, provides a novel internal mechanism for understanding the relationship between total rewards and team member proactivity.

Lastly, the influence of perceived corporate social responsibility (CSR) broadens the boundary conditions of total rewards on team member proactivity. Previous research predominantly focused on the relationship between rewards and incentives and employee behavior within organizations or teams. However, as the new generation of employees increasingly values corporate reputation, the interplay between internal and external reputations can significantly motivate them. Consequently, this study highlights the importance of CSR as a regulatory condition that the new generation of employees should consider when shaping the overall rewards system. Drawing on social identity theory, this study perceives CSR as a tool for enterprises to enhance their identity value. By examining the impact of team member proactivity from a process perspective, wherein the interaction between these factors influences an individual’s overall motivation, the study offers a new and robust case for social identity theory.

### 5.2. Practical Implications

First, enterprises should establish the incentive concept of total rewards rather than simply using a single economic reward or only one or two kinds of non-economic rewards. Instead, they should comprehensively understand the needs and dynamic changes of the new generation of employees in various ways and then optimize the comprehensive design scheme of total rewards based on this understanding to achieve effective management and employee incentives. Enterprises can design total rewards schemes to incentivize team participation, such as rewarding employees who contribute to team efforts or assigning them key roles within the team. This approach aims to encourage greater team member proactivity, facilitate team adaptation, and enhance overall team performance.

Second, managers should attach importance to the cultivation of the new generation of employees’ calling and fully stimulate their sense of calling effectiveness. Calling is not fixed after early formation but can be actively intervened with and dynamically changed after entering the career field [20]. Through leadership behavior, organizational culture training, employee psychological construction, role model publicity, and other ways, strengthen the construction of work meaning, effectively transform the supportive environment of the organization’s total rewards into employees’ pride and sense of meaning, enhance employees’ sense of responsibility for work, team, and enterprise, and then stimulate, awaken, or strengthen the calling of the new generation of employees.

Lastly, businesses should proactively engage in additional corporate social responsibility initiatives, incorporating them into their overall corporate strategy and human resource management practices. They should also encourage and guide their younger employees to actively participate in CSR activities, ensuring effective communication and timely feedback. Furthermore, they should integrate the concept of CSR into specific tasks to reinforce altruistic values and foster a stronger sense of organizational identity.

In contrast to the above discussion, some argue that the traditional view of CSR does not fully account for the diversity of industries. Recently, CSR research has focused more on “controversial industries”, which are characterized by significant social or environmental impacts and can provoke strong negative reactions, such as oil, tobacco, gambling, and weapons. Stakeholder responses to CSR in these industries vary widely and are often negative, as these industries are seen as having inherently negative attributes. For example, Abdulalem Mohammed revealed that CSR related to the environment has significant direct and indirect impacts on customer loyalty [55]. This perception can lead to a backlash, increasing company risk and triggering critical events like protests or boycotts, ultimately damaging the company’s reputation.

### 5.3. Limits of Research and Perspectives

This study may have potential limitations that future research should address. Firstly, while this paper treats total rewards as a comprehensive academic construct encompassing various incentives, it possesses rich connotations and a multidimensional structure. Future studies could use methods like Qualitative Comparative Analysis (QCA) to investigate the complementary and synergistic effects of different reward configurations on the proactive behaviors of the new generation of employees, as well as within specific subgroups such as different industries and stages of enterprise development. Secondly, future research could delve deeper into how other traits, values, and situational factors of the new generation of employees influence the development of team member proactivity. In addition to these limits, we conducted our tests on a sample of mainland Chinese firms and post-1990s Chinese employees. Despite the representativeness of our sampling on Chinese multi-industry and multi-field firms, future scholars could improve the generalizability of our findings to other national contexts. In addition to these limitations, the research group in this paper is part of the conventional industry and sector rather than a controversial one [56]. Therefore, it is necessary to further investigate and discuss whether the research conclusions, particularly the finding that the perception of corporate social responsibility has a regulating effect across all industries and organizations, are universally applicable. Furthermore, the interplay between internal and external reputations is crucial for motivating employees, with CSR often enhancing external reputation. However, the perceived value of CSR varies among organizations, with some brands not viewing it as beneficial or aligned with their values. This perception influences their investment in CSR activities and integration into broader strategies. Future research should explore these differences and their impact on internal motivation and external reputation.

## 6. Conclusions

Focusing on the new generation of employees born after 1990 in China, this study constructs a moderated mediation model based on social identity theory, which offers a robust explanatory framework for the impact of total rewards on team member proactivity. The study highlighted several insights. Firstly, total rewards significantly enhance the proactivity of team members among the new generation of employees. Furthermore, the total rewards foster team member proactivity by boosting employees’ sense of calling. Lastly, when employees have a strong perception of corporate social responsibility, the positive influence of total rewards on calling is amplified, and the indirect effect of total rewards on enhancing team member proactivity through the stimulation of calling is also strengthened. In a nutshell, this research provides valuable insights for managers aiming to effectively engage the new generation of employees and boost team performance. In essence, our model enriches the understanding of how compensation practices can be leveraged to boost proactivity among the new generation of employees.

## Figures and Tables

**Figure 1 behavsci-14-00670-f001:**
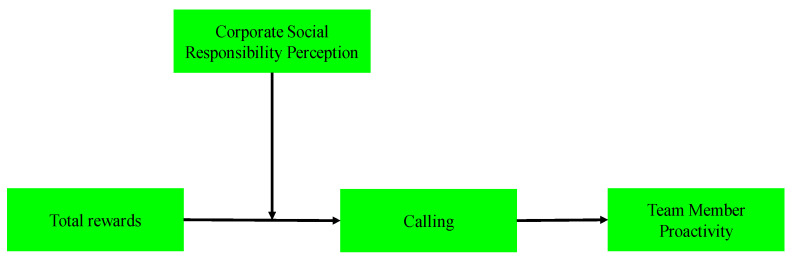
Conceptual model.

**Figure 2 behavsci-14-00670-f002:**
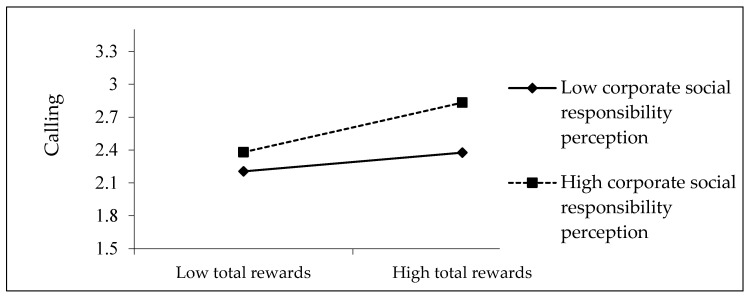
The moderating effect of CSR perception on the relationship between total rewards and calling.

**Table 1 behavsci-14-00670-t001:** Repartition of different generations.

Generation Class	Frequency (%)
Post-1990	43.11
Post-1995	51.71
Post-2000	5.18
Total	100

**Table 2 behavsci-14-00670-t002:** Repartition of the respondent according to their work experience.

Work Experience (Years)	Frequency (%)
Less than 1	22.87
1 to 3	31.1
3 to 5	12.8
5 to 10	17.68
More than 10	15.55
Total	100

**Table 3 behavsci-14-00670-t003:** Measurement model comparisons.

Model	*χ* ^2^	*df*	*χ*^2^/*df*	*RMSEA*	*CFI*	*TLI*	*IFI*
Four-factor model (*TR*, *CSRP*, *C*, *TP*)	3172.068	1945	1.631	0.044	0.915	0.909	0.916
Three-factor model (*TR*, *CSRP*, *C + TP*)	4952.035	1989	2.490	0.067	0.795	0.785	0.796
Three-factor model (*TR*, *CSRP + C*, *TP*)	5137.333	1987	2.585	0.070	0.782	0.772	0.783
Two-factor model (*TR + CSRP + C*, *TP*)	5468.115	1991	2.746	0.073	0.759	0.748	0.761
Two-factor model (*TR*, *CSRP + C+TP*)	5804.997	1991	2.916	0.077	0.736	0.724	0.738
One-factor model (*TR + CSRP + C+TP*)	6421.008	1993	3.222	0.082	0.693	0.680	0.695

Note: N = 328; *TR* = total rewards, *C* = calling, *CSRP*= CSR perception, *TP* = team member proactivity.

**Table 4 behavsci-14-00670-t004:** Mean, standard deviation (SD), and correlation coefficient.

N	Variable	Mean	SD	1	2	3
1	*TR*	4.580	0.620	1		
2	Cal	3.970	0.710	0.398 **	1	
3	*CSRP*	4.350	0.730	0.517 **	0.381 **	1
4	*TP*	3.960	0.890	0.207 **	0.212 **	0.196 **

Note: *TR* = total rewards, Cal = calling, *CSRP* = CSR perception, *TP* = team member proactivity. ** *p* < 0.01.

**Table 5 behavsci-14-00670-t005:** Hierarchical regression of the whole model.

Variable	Calling				Team Member Proactivity			
	M1	M2	M3	M4	M5	M6	M7	M8
1. Sex	0.149 *	0.102	0.102	0.108	0.206 *	0.175	0.158	0.174
2. Age	0.009	0.013	0.012	0.011	−0.001	0.002	0.000	−0.004
3. Education	−0.072 *	−0.053	−0.042	−0.033	0.009	0.021	0.03	0.025
4. Seniority	0.007	0.005	0.003	0.004	0.043 **	0.042 **	0.041 **	0.04 **
5. Grade	0.184 *	0.154 *	0.143	0.136	0.137	0.117	0.092	0.098
6. Working time	−0.024	−0.025	−0.02	−0.017	−0.012	−0.012	−0.008	−0.006
7. Superior and subordinate relationship (T2)	0.263 ***	0.222 ***	0.215 ***	0.213 ***	0.042	0.014	−0.022	−0.013
8. Proactive personality (T1)	0.226 ***	0.118 *	0.08	0.074	0.199 **	0.127	0.107	0.136
9. *TR* (T1)		0.358 ***	0.251 ***	0.22 ***		0.24 **	0.181 *	
10. *C* (T2)							0.165 *	0.218 **
11. *CSRP* (T1)			0.216 ***	0.208 ***				
12. *TR***CSRP*			0.154 *	0.155 *				
R^2^	0.176	0.259	0.300	0.325	0.084	0.107	0.120	0.127
ΔR^2^	0.176	0.084	0.010	0.018	0.084	0.024	0.013	0.013
F	8.490 ***	12.371 ***	12.309 ***	11.608 ***	3.633 ***	4.247 ***	4.325 **	4.198 ***

Note: T1 = Time 1, T2 = Time 2. * *p* < 0.05, ** *p* < 0.01, *** *p* < 0.001.

**Table 6 behavsci-14-00670-t006:** Moderated mediating effect test results.

Process Macro	Regulating Variable	Path	B	SE	95% Confidence Interval
M7	CSRP	Low (−1 SD)	0.023	0.021	[−0.001, 0.088]
		High (+1 SD)	0.060	0.035	[0.004, 0.142]

## Data Availability

The data presented in this study are available on request from the corresponding author.

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
