# Peer review of "Addressing the “Lying Flat” Challenge in China: Incentive Mechanisms for New-Generation Employees through a Moderated Mediation Model"

_behavsci, 2024, doi:10.3390/bs14080670_

Round 1

Reviewer 1 Report

Comments and Suggestions for Authors

Thank you for the opportunity to read and review the manuscript: “Addressing the ‘Lying Flat’ Challenge in China: Incentive Mechanisms for New Generation Employees through a Moderated Mediation Model.”

Overall, it is a very well-written manuscript with well-developed arguments and hypotheses. However, I have one observation: CSR was reported to act as the moderating variable. This needs to be clearly justified, especially when you are focusing on calling as the mediator for proactivity.

In the theoretical contribution section, it is stated: “both theoretical and empirical research on total rewards remains lacking.” Can you please elaborate on what you mean by this being ignored and why it is important? Similarly, the concept of employees' proactive behavior should be explained further.

In the same section, you discuss the interplay between internal and external reputations that can significantly motivate employees and mention CSR as influencing the latter. One possible area to explore is the variation in how different organizations perceive CSR, as for some brands and organizations, CSR might not necessarily be considered beneficial.

Good luck with the revisions!

Author Response

Dear review,

Thank you for your time and helpful suggestions. Your comments were beneficial in improving our article as they were interesting. Additionally, we have gained tremendous experience during this editing process. These skills will come in very handy in the future. All of your recommendations have been carefully considered and included in the article.

Consequently, we proceeded step by step to make the correction clear and readable. The steps are described below.

  • We started by reading all of the reviews' suggestions and recommendations. After that, we prepared a draft to enhance the literature by considering the reviewers' suggestions.
  • Following a comprehensive examination of the draft, we used "Track Changes" to include each comment and suggestion in the paper one by one. After this, we get through the spelling, grammar, and sentence structure.
  • In the final stage, we submitted the manuscript in two native languages to improve the written English.
  • The article is finally checked and submitted.

We confirm that this manuscript has not been published elsewhere and is not being considered by another journal. All authors have approved the manuscript and agree with its submission to this section. Please address all correspondence concerning this manuscript to us at [email protected]

With my best regards,

Bonoua FAYE

Response to Reviewer 1

Thank you for the opportunity to read and review the manuscript: “Addressing the ‘Lying Flat’ Challenge in China: Incentive Mechanisms for New Generation Employees through a Moderated Mediation Model.”

Comment 1: Overall, it is a very well-written manuscript with well-developed arguments and hypotheses. However, I have one observation: CSR was reported to act as the moderating variable. This needs to be clearly justified, especially when you are focusing on calling as the mediator for proactivity.

Response 1: This section has been improved accordingly. Kindly see the comment on page 7, from lines 344 to 365.

Comment 2: In the theoretical contribution section, it is stated: “Both theoretical and empirical research on total rewards remains lacking.” Can you please elaborate on what you mean by this being ignored and why it is important? Similarly, the concept of employees' proactive behavior should be explained further.

Response 2: This paragraph has been improved. Kindly see page 13, from lines 567 to 582.

Comment 3: In the same section, you discuss the interplay between internal and external reputations that can significantly motivate employees and mention CSR as influencing the latter. One possible area to explore is the variation in how different organizations perceive CSR, as for some brands and organizations, CSR might not necessarily be considered beneficial.

Response 3: Because this paper does not specifically discuss the varying perceived value of CSR among organizations, we suggest this as an area for further research. This topic could be integrated into the limitations and future research section. Please refer to page 15, lines 670 to 681. However, we have also improved the discussion section. Kindly see lines 647-656.

Best regards

Reviewer 2 Report

Comments and Suggestions for Authors

Thank you for the opportunity to read your manuscript, which I found to be well-written, theoretically grounded, and logically cohesive. It reminded me of a classic article by Christopher Earley that was published on 1989 in Administrative Science Quarterly on social loafing. The study found that collectivists in China were less likely to loaf in groups than individualists in the U.S. because they would lose face. In this study, you're using a newer term, which the authors say is popular in China: "lying flat." It would be interesting to add the latter concept (losing face) into this study and consider why younger generations may be comfortable "lying flat." 

The findings on whether proactive team behaviors are related to perceptions of corporate social responsibility, calling, and total rewards are interesting, yet not counterintuitive. Even so, the findings should be useful to organizations as they consider the importance of total reward packages and calling. I found your methodology to be robust as you included multiple methods and models, including bootstrapping. I also appreciate your inclusion of both theoretical and practical implications.  

At one point in the paper, you reference "expectation" theory but I believe that should be "expectancy" theory. Also, it would be helpful if you gave an example of a case that you removed from the dataset because it lacked logical reasoning.

Author Response

Dear review,

Thank you for your time and helpful suggestions. Your comments were beneficial in improving our article as they were interesting. Additionally, we have gained tremendous experience during this editing process. These skills will come in very handy in the future. All of your recommendations have been carefully considered and included in the article.

Consequently, we proceeded step by step to make the correction clear and readable. The steps are described below.

  • We started by reading all of the reviews' suggestions and recommendations. After that, we prepared a draft to enhance the literature by considering the reviewers' suggestions.
  • Following a comprehensive examination of the draft, we used "Track Changes" to include each comment and suggestion in the paper one by one. After this, we get through the spelling, grammar, and sentence structure.
  • In the final stage, we submitted the manuscript in two native languages to improve the written English.
  • The article is finally checked and submitted.

We confirm that this manuscript has not been published elsewhere and is not being considered by another journal. All authors have approved the manuscript and agree with its submission to this section. Please address all correspondence concerning this manuscript to us at [email protected]

With my best regards,

Bonoua FAYE

Response to Reviewer 2

Comment 1: Thank you for the opportunity to read your manuscript, which I found to be well-written, theoretically grounded, and logically cohesive. It reminded me of a classic article by Christopher Earley that was published in 1989 in Administrative Science Quarterly on social loafing. The study found that collectivists in China were less likely to loaf in groups than individualists in the U.S. because they would lose face. In this study, you're using a newer term, which the authors say is popular in China: "lying flat." It would be interesting to add the latter concept (losing face) into this study and consider why younger generations may be comfortable "lying flat."

Response 1: Thank you for this suggestion. We have read that paper and gained Knowledge. Our study closely focuses on the concept of “lying flat.” However, for the audience, the concept of “losing face” has been briefly introduced. Kindly see page 2, lines 84-90.

Comment 2: The findings on whether proactive team behaviors are related to perceptions of corporate social responsibility, calling, and total rewards are interesting, yet not counterintuitive. Even so, the findings should be useful to organizations as they consider the importance of total reward packages and calling. I found your methodology to be robust as you included multiple methods and models, including bootstrapping. I also appreciate your inclusion of both theoretical and practical implications. 

Response 2: Thank you again for your effort to improve our manuscript.

Comment 3: At one point in the paper, you reference "expectation" theory, but I believe that should be "expectancy" theory. Also, it would be helpful if you gave an example of a case that you removed from the dataset because it lacked logical reasoning.

Response 3: The mistake has been corrected. Kindly see line 187 on page 4.

Best regards.